# Polyphenol-Rich Beverage Consumption Affecting Parameters of the Lipid Metabolism in Healthy Subjects

**DOI:** 10.3390/ijms24010841

**Published:** 2023-01-03

**Authors:** Celina Rahn, Tamara Bakuradze, Simone Stegmüller, Jens Galan, Sonja Niesen, Peter Winterhalter, Elke Richling

**Affiliations:** 1Division of Food Chemistry and Toxicology, Department of Chemistry, Rheinland-Pfälzische Technische Universität Kaiserslautern-Landau, Erwin-Schrödinger-Straße 52, D-67663 Kaiserslautern, Germany; 2Medical Institute, Hochgewanne 19, D-67269 Grünstadt, Germany; 3Institute of Food Chemistry, Technische Universität Braunschweig, Schleinitzstraße 20, D-38106 Braunschweig, Germany

**Keywords:** human intervention study, polyphenol-rich beverage, lipid metabolism, phosphodiesterase activity, DNA integrity

## Abstract

Polyphenols are a diverse and widely distributed class of secondary metabolites, which possess numerous beneficial properties including a modulation of glucose and lipid metabolism. This placebo-controlled human intervention study was performed to explore effects of polyphenol-rich beverage (PRB) uptake on lipid metabolism, as well as DNA integrity. In this case, 36 healthy men were randomly divided to consume either 750 mL of a PRB (containing 51% chokeberry, cranberry, and pomegranate) or a placebo drink daily for eight weeks. Only PRB consumption was found to decrease fat and protein intakes significantly compared to the preceding one-week washout period. During the intervention with PRB an increased fat-free mass was shown after four weeks, whereas a significant elevation in body weight and leptin was observed in placebo group. Blood lipids were not significantly altered after PRB consumption, while triglyceride levels increased after placebo drink intake. In platelets, a significant inhibition of phosphodiesterase (PDE) activity was observed, more pronounced in test group. Consuming the PRB decreased total DNA strand breaks in whole blood as well as H_2_O_2_-induced breaks in isolated lymphocytes. Overall, our study suggested beneficial effects on lipid metabolism by reduced energy intake, modulation of biomarkers such as PDE activity and improved DNA integrity associated with PRB consumption.

## 1. Introduction

Overweight and obesity are characterized by excessive fat accumulation and classified by a body mass index (BMI) greater than or equal to 25 or 30 kg m^−2^, respectively. Since the prevalence of overweight and obesity in humans has risen in recent years and increased the risk for diseases such as type 2 diabetes, cardiovascular diseases, and cancer, it has become a major health problem worldwide [1]. Growing evidence suggests that polyphenols could have protective effect against these disorders.

Polyphenols are a relevant class of secondary metabolites, which show highly diverse structures and consist of many subclasses such as flavonoids (e.g., anthocyanins), phenolic acids, stilbenes, or lignans [2]. They are abundant in the human diet, particularly in fruits, vegetables, beverages or cereals, and the total polyphenol contents ranges from more than 15,000 mg per 100 g in cloves, 1432 mg per 100 g in black chokeberry to 7.8 mg per 100 mL in rosé wine [3]. Many beneficial effects have been described for polyphenols in vitro and in vivo. They possess anti-bacterial or anti-inflammatory properties and can affect glucose metabolism as well as gut microbiota [4,5,6,7,8]. Furthermore, many studies also highlighted DNA-protective and antioxidant effects in humans [5,8,9]. Additionally, polyphenol-rich products are discussed in the context of influencing lipid metabolism by lowering body weight, improvement of plasma lipid levels or change of lipid profile [10,11,12,13,14]. Several studies have demonstrated that especially anthocyanin-rich red fruits and their juices are associated with health benefits in various experimental models. It has been reported that the daily consumption of anthocyanin-rich fruit juice was linked with a significant reduction in body fat, an increase of fat-free mass (FFM) and it can affect blood lipid profiles by decreasing triglyceride (TG), total cholesterol (TC), and low-density lipoprotein (LDL) cholesterol concentrations [9,14,15]. Thus, weight loss and maintaining a healthy body weight by reduced energy intake and an increased consumption of fruits and vegetables has also been indicated to effectively lower the risk for morbidities associated with obesity [1,16,17]. A proposed mechanism for weight loss is the stimulation of lipolysis—the hydrolyzation of TGs to free fatty acids and glycerol—via inhibition of 3′,5′-cyclic adenosine monophosphate (cAMP)-phosphodiesterase (PDE), resulting in increased cAMP levels [18]. Polyphenols and foods containing flavonoids have shown PDE-inhibiting effects in vitro as well as in vivo [19,20,21,22,23]. Furthermore, many studies evidenced that plasma leptin levels are correlated with the body weight and can be a useful bio-marker for obesity, since the levels increase with overweight and decrease with weight loss [24]. Foods rich in anthocyanins have also demonstrated to significantly reduce body weight or BMI, and decrease leptin blood levels as shown by supplementation with plum juice in a human intervention study [11]. Consequently, the presence of the metabolic syndrome is associated with statistically significant higher values of plasma and serum leptin [25,26]. Furthermore, uric acid, an end product of the purine nucleotide metabolism in humans, has received attention recently. Elevated plasma uric acid levels are a risk factor of both, gout and obesity, which may result in an increased incidence of the metabolic syndrome, characterized by visceral obesity, dyslipidemia, hyperglycemia, and hypertension [27,28].

A previous intervention study found that the consumption of 750 mL anthocyanin-rich fruit juice modulated body composition in healthy male volunteers [9]. Therefore, this placebo-controlled human intervention study reported here focuses on the influence of polyphenol-rich beverage (PRB) on lipid metabolism. Preliminary findings on lipid metabolism in vitro regarding lipid accumulation, lipolysis, and PDE activity showed that extracts of the red berries chokeberry, cranberry, and pomegranate revealed the most potent effects [29]. Based on these in vitro results, the beverage tested here was produced from chokeberry, cranberry, and pomegranate. The different biomarkers of the lipid metabolism such as blood lipids, hunger and satiety hormones as well as PDE activity in platelets were analyzed after daily consumption of 750 mL PRB over an eight-week period. In addition, the effects on anthropometric measurements and food intake were determined. Moreover, plasma uric acid concentrations, background, and total DNA strand breaks in whole blood as well as H_2_O_2_-induced DNA strand breaks in isolated peripheral blood lymphocytes (PBLs) were examined.

## 2. Results

### 2.1. Composition of the Study Beverages

The composition of the PRB and placebo drink is summarized in Table 1. Due to the comparability of the two study beverages, the participants received isocaloric beverages, each providing 315.1 kcal L^−1^. Both study beverages had similar Brix values, myo-inositol contents as well as sugars and identical acid concentrations, but differed in their sorbitol (8.7 g L^−1^ vs. 0 g L^−1^) and polyphenol contents (3.0 g gallic acid equivalents (GAE) L^−1^ vs. 0.1 g L^−1^). Anthocyanins determined were cyanidin-derivative I (4.6%) and II (2.4%), cyanidin-3-galactoside (60.3%), cyanidin-3-glucoside (5.5%), cyanidin-3-arabinoside (22.2%), peonidin-3-galactoside (1.1%), and cyanidin-3-xyloside (3.9%) with a total of 735.8 mg L^−1^. Flavonoids identified were myricetin-hexoside (3.8%), quercetin-hexoside-pentoside (6.7%), syringetin-hexoside-derivative (27.0%), quercetin-rhamnoside-hexoside I (7.9%) and II (8.2%), quercetin-3-galactoside (21.2%), quercetin-3-glucoside (11.5%), and quercetin (13.6%) with a total amount of 235.0 mg L^−1^. High contents of the chlorogenic acids, i.e., neochlorogenic acid (3-CQA, 54.6%) and chlorogenic acid (5-CQA, 45.4%) with a sum of 306.9 mg L^−1^ as well as tannins (α-punicalin (25.2%), β-punicalin (38.7%), punicalagin-derivatives I (1.8%) and II (4.3%), α-punicalagin (8.6%), and β-punicalagin (21.4%)) with a total amount of 400.8 mg L^−1^ were found in the PRB compared to the placebo drink.

### 2.2. Compliance

Compliance was monitored on the one hand by seven-day food records, which showed that the participants complied with the dietary restrictions. On the other hand, spot urine samples from the participants were analyzed for anthocyanins by high-performance liquid chromatography/electrospray ionization tandem mass spectrometry (HPLC-ESI-MS/MS) previously reported by Mueller et al. with slight modifications [30]. The monitoring of anthocyanins in the urine of participants from both groups showed levels below the LOD after the washout period. During the intervention phase, urine samples of the test group demonstrated detectable concentrations of anthocyanins, while no anthocyanins were detected in urine samples of the placebo group (data not shown).

### 2.3. Nutrient Intake

The daily energy and nutrient intakes of the volunteers were calculated based on seven-day food records, which were completed at the end of the washout period as well as after four and eight weeks of intervention (Table 2). Although the participants consumed equicaloric study beverages, significantly decreased protein intakes after four weeks as well as decreased fat and protein intakes after eight weeks of intervention in comparison to the washout period were found only in the test group. In contrast, consumption of placebo drink did not influence noticeably any of the measured nutrient intake parameters. Test group and placebo group differed significantly (*p* < 0.05) in terms of fat intake after eight weeks of the intervention.

### 2.4. Body Weight and Body Composition

Anthropometric measurements were carried out after the washout period as well as after four and eight weeks of intervention (Table 3). In Figure 1 the changes of body weight, fat-free mass (FFM), and fat mass (FM) in both groups during the intervention compared to the end of the washout period are presented. Both groups of participants showed similar trends in body weight and composition over the time course of the study. Notably, there was a decrease in FM and significant increase in FFM after four weeks consumption of the PRB, whereas a significant elevation in body weight was observed in the placebo group. Further uptake of both study drinks showed no significant modulation of these parameters during the following weeks. The two groups did not significantly differ in any of the measured parameter of body weight and composition, respectively.

### 2.5. Blood Lipid Profile and Lipase Activity

In Table 4 the results from analyses of blood lipids and lipase activity are summarized. No modulation of neither blood lipids such as TGs or high-density lipoprotein (HDL) cholesterol nor lipase activity was detected in the test group throughout the entire study period. Compared to the test group, consumption of placebo drink led to significantly increased TG levels during intervention. Lipase activity was slightly increased in placebo group during the whole intervention period.

### 2.6. Plasma Concentrations of Leptin and Glucagon-Like Peptide-1 (GLP-1)

The equation of energy intake and expenditure is tightly controlled by hormonal signaling among the gut, brain, and adipose tissue [31]. Table 5 provides the concentrations of the two hormones leptin and glucagon-like peptide-1 (GLP-1). No significant differences in leptin plasma concentrations after the consumption of PRB compared to the preceding washout period were observed. However, a slight decrease in leptin concentrations was observed in the test group after four weeks of intervention. The consumption of placebo drink resulted in a significantly higher leptin plasma concentrations compared to the preceding washout period. In general, the placebo group showed higher baseline levels of leptin compared to the test group. In contrast, administration of placebo drink slightly increased GLP-1 levels, but the changes were not significant (Table 5), whereas in test group a significant decrease in GLP-1 compared to the washout period was shown.

### 2.7. Platelet cAMP-PDE Activity

Platelets possess three PDE isoforms (PDE 2, PDE 3, and PDE 5), which catalyze the hydrolysis of the second messengers cAMP and cyclic guanosine monophosphate, and thus, play an important role in the regulation of lipid metabolism [32]. In order to confirm the inhibitory effects of polyphenol-rich fruit juice and concentrate extracts as well as their fractions on PDE 3B activity in vitro, which we had observed in our previous studies [21,29], we investigated the effects of PRB uptake on cAMP-PDE activity in platelets in this human intervention study. The results of cAMP-PDE activity are presented in Figure 2. Compared to baseline, consumption of the study beverages significantly reduced cAMP-PDE activity after four- and eight-week intervention in both groups. Surprisingly, participants who consumed the PRB experienced a stronger (but not significant) decrease compared to placebo group.

### 2.8. Uric Acid

Considering the relationships between uric acid levels and obesity or metabolic syndrome [27,28], respectively, we also determined the uric acid concentrations during the study using HPLC-ESI-MS/MS (Figure 3). The levels of plasma uric acid were all within the normal range at baseline. After consumption of PRB for four and eight weeks, uric acid levels decreased slightly. A similar trend in uric acid levels was observed after intervention of placebo drink, but with a more pronounced and significant decrease after eight weeks compared to the washout period. However, the two groups did not significantly differ in terms of plasma uric acid concentrations at any point of the study period.

### 2.9. DNA Strand Breaks (Comet Assay)

The modulation of background and total DNA strand breaks was assessed in whole blood. Additionally, to monitor alterations in cell sensitivity to reactive oxygen species-induced DNA damage, the total DNA strand breaks in PBLs were analyzed after H_2_O_2_ challenge. The results of background and total DNA strand breaks in whole blood are presented in Figure 4 and showed a significant decrease of total DNA strand breaks (with formamidopyrimidine-DNA glycosylase (FPG) treatment) in test group during the whole study period, whereas no modulation in background DNA breaks (without FPG treatment) were observed. In placebo group DNA strand breaks remained unchanged except a slight significant reduction in background DNA breaks after eight weeks compared to washout period.

The results of H_2_O_2_-induced total DNA strand breaks are presented in Figure 5. In test group a distinct (but not significant) decrease of DNA damage was observed, whereas no modulation was demonstrated in placebo group during the intervention period. However, a significant difference was observed between the test and placebo groups after eight weeks of the intervention.

### 2.10. Correlation Analysis

In this human intervention study 20 parameters on nutrient intake, body composition, blood lipids, hunger and satiety hormones, enzymes of the lipid metabolism, DNA integrity, and uric acid were investigated. In order to receive insights into relationships between various pairs of parameters, we performed Spearman correlations analysis. The heat maps (Figure 6 and Figure 7) show the calculated correlation coefficients (rho) of all parameters determined in this study. Spearman correlation coefficients were calculated based on the changes of the washout period to the intervention period after four weeks. There was a weak correlation between energy and nutrient intake and body composition (FM and FFM; rho = −0.28 and 0.17). A positive and partly significant correlation was shown between leptin and body weight, BMI, and FM (rho = 0.25 to 0.42). Likewise, a significant positive correlation was obtained for TG levels with body weight and BMI, respectively (rho = 0.38 and 0.43; *p* < 0.02). Total DNA strand breaks in PBLs correlated negatively with HDL cholesterol (rho = −0.33; *p* = 0.05). Furthermore, the expected correlations of the individual parameters within the group of energy and nutrient intake as well as within the group of body composition could be confirmed, such as the significant correlation of body weight with FM (rho = 0.37; *p* = 0.03) or the significant correlations of energy intake with protein, fat, and carbohydrate intake (rho = 0.56 to 0.79; *p* < 0.001).

A similar trend was shown in the correlation analysis based on the changes of the washout period to the intervention period after eight weeks (Figure 7). There was a weak positive correlation between FM and protein intake as well as FM and fat intake (rho = 0.21 and 0.15). A negative correlation was found for FFM with energy intake and with nutrient intake (rho = −0.30 to −0.09). For protein intake and TC levels a significant positive correlation was observed (rho = 0.41; *p* = 0.01). Similar to the effects observed after four weeks, leptin was significantly correlated with body weight as well as BMI after eight weeks (rho = 0.41 and 0.45, respectively; *p* ≤ 0.01). Further correlations were shown between lipase activity and FM (rho = 0.33; *p* = 0.05) as well as for total DNA strand breaks in whole blood (with FPG) and uric acid content (rho = 0.41; *p* = 0.01).

## 3. Discussion

It is well known that polyphenols possess a plethora of beneficial properties. The aim of the present human intervention study was to investigate the effects of PRB consumption on different biomarkers regarding lipid metabolism, such as blood lipids or PDE activity. In addition, nutrient intake, hunger, and satiety hormones as well as uric acid concentrations were monitored. Supplementary, parameters of oxidative stress such as DNA strand breaks were examined. The participants consumed either 750 mL of the PRB on a daily basis over an eight-week intervention period, which was produced from different red fruit juices/concentrates (chokeberry, cranberry, and pomegranate) or 750 mL of the placebo drink. These three different fruit varieties were chosen here based on preliminary in vitro studies on lipid accumulation, lipolysis, and PDE activity [21,29]. Study PRB as well as placebo drink were produced isocaloric but differed slightly in their fructose content (33.3 g L^−1^ vs. 38.6 g L^−1^). Fructose can have a lipogenic effect by stimulating the fatty acid synthesis [33]. Additionally, the PRB contained relatively high amounts of polyphenols (3.0 g GAE L^−1^) and total anthocyanins (735.8 mg L^−1^) as well as flavonoids (235.0 mg L^−1^), chlorogenic acids (306.9 mg L^−1^), and tannins (400.8 mg L^−1^). The high concentration of sorbitol determined in PRB as a natural ingredient of chokeberry. Since no polyphenols or natural ingredients of red fruits such as sorbitol were present in the placebo drink, urine was checked for anthocyanins as compliance marker. The obtained results approved that all participants were compliant and had adhered to the study protocol.

Evaluation of the food and beverage records pointed out that only PRB consumption was found to decrease fat and protein intakes significantly compared to the washout period. No modulation in carbohydrate intake might be observed, since the PRB consumed during the intervention period was rich in carbohydrates, especially glucose and fructose. Although the participants consumed isocaloric study beverages, energy and nutrient intake did not alter in placebo group after uptake of placebo drink. The trend we identified in the present study towards reducing energy intake, caused by the consumption of the polyphenol-rich drink, might help to improve the satiating capacity and, thus, support body weight management. In C57BL/6J mice a decreased food intake in proportion to the concentration of cyanidin-3-galactoside enriched chokeberry extract treatment was found [34]. In line with our results, in a human intervention study with 57 healthy male volunteers, Bakuradze et al. observed a significantly increased energy and carbohydrate intake but decreased fat and protein intake after four and eight weeks of anthocyanin-rich fruit juice consumption in comparison to the washout period [9]. Furthermore, in overweight but otherwise healthy men, daily energy intake was significantly lower at week 11 after intervention of orange juice (250 mL/d) when compared to pre-intervention [35].

In the current study, we provide evidence of the beneficial effects of PRB consumption (containing chokeberry, cranberry, and pomegranate juice; total phenolics = 3.0 g L^−1^) on anthropometric parameters. During the intervention with PRB, a significant increased FFM was shown after four weeks, while a significant elevation in body weight was observed after placebo drink consumption. Numerous studies investigated the effects of polyphenols on body composition. In agreement with our findings, Bakuradze and co-workers reported that male volunteers who consumed daily 750 mL of an anthocyanin-rich fruit juice (total phenolics = 3.6 g L^−1^) experienced a significant increase in FFM and a significant decrease of FM after one and four week intervention [9]. Similar effects on body composition were observed in a study conducted with 35 elderly women by Costa et al. They showed a significantly reduced FM and BMI after grape juice consumption (400 mL d^−1^) [10]. Other data highlighted that plum juice supplementation (200 mL d^−1^) over four weeks significantly reduced body weight and BMI in 20 healthy volunteers [11]. In a prospective cohort study with 54,787 Danish men and women, it was observed that higher flavonoid intakes were cross-sectionally associated with lower estimates of body fat [36].

Regarding the impact of PRB consumption on the regulation of lipid metabolism in terms of hormonal signaling, GLP-1 and leptin play a central role. GLP-1 is secreted by pancreatic cells and is involved in the reducing gastric emptying and food intake, thus promoting satiety but also in lowering blood glucose [37]. In the present study, we analyzed slightly increased GLP-1 levels after placebo drink consumption. However, with PRB intervention in healthy men a slight increase in GLP-1 was observed after four weeks, whereas a significant decrease in GLP-1 levels was shown after eight weeks compared to the washout period. Many of the studies investigated the short-term effects of GLP-1 or examined the effects in subjects with type 2 diabetes [38,39]. However, in healthy male C57BL/6J mice fed with a diet including 40 mg anthocyanidins/kg body weight for 14 weeks, Daveri et al. observed significantly higher plasma GLP-1 concentration compared to control group [40]. In line with these results, delphinidin-3-rutinoside significantly increased the secretion of GLP-1 in GLUTag cells, followed by delphinidin and malvidin [41]. The murine GLUTag cell line is widely used to investigate the effects of food components on GLP-1 secretion and to examine the signaling pathways underlying nutrient-induced GLP-1 release [42]. Controversially, Akbarpour and co-workers described no significant changes in GLP-1 after 100 mL intervention of pomegranate juice daily over eight weeks in women with type 2 diabetes [43]. In addition, another study showed that plasma concentrations of GLP-1 were rapidly increased in response to ileal carbohydrate and lipid perfusion [44]. The reduction in GLP-1 levels we observed after eight weeks intervention of PRB might be linked to the reduced fat intake in the test group. The type of test system, study subjects (healthy or not), the consumed product, inter-individual variation or other differences in study conditions may be responsible for these conflicting results. Still, we cannot fully explain the reduction in GLP-1 after eight weeks PRB consumption in our study. Future research is necessary to understand this effect. Another hormone, namely leptin, is important for regulating energy balance. It is produced by the adipose tissue in proportion to the size of fat stores and secreted by adipocytes. The receptors for leptin are mainly found in the hypothalamus, which is known to act in controlling food intake and metabolic rate. Normally, leptin concentrations correlate with body weight and FM, so the levels are elevated in overweight and obese people [45,46]. We observed significantly higher leptin concentrations compared to the preceding washout period in placebo group after the whole intervention period, whereas consumption of PRB resulted in a slight decrease in leptin plasma concentrations after four weeks. Likewise, Tucakovic et al. described a decrease (but not significant) in leptin levels after four weeks of intervention with 200 mL anthocyanin-rich plum juice in 20 healthy individuals [11]. Previously studies with animals have shown similar findings where cyanidin-3-glucoside and plum juice decreased plasma leptin in Wistar rats or grape-bilberry juice reduced serum leptin in Fischer rats [47,48]. We hypothesized that we received only indications of hormonal effects in our study since the number of participants was small (n = 18 per group) and we found a high inter-individual variation in leptin and GLP-1 levels as it was shown by high differences in baseline levels between the participants. Several studies have discussed the impact of gender and age on hormone status [49,50,51]. Since we had only young male volunteers participating in the study, these two factors are of minor importance. Furthermore, environmental factors, genetic variability in receptors, emotional states or psychological variables can affect hormone levels, too, which can result in a high inter-individual variation [52,53]. An additional factor influencing hormone concentrations might be the inter-individual variation in the absorption, distribution, and elimination of polyphenols, which may be down to the variation of the volunteer’s gut microbiota [54,55,56]. Despite the variability, we confirmed the significant, positive relationship between leptin and body weight as well as BMI within the framework of the correlation analysis (see Section 2.10). In line with our findings, Rangel-Huerta et al. observed a significant, positive correlation between BMI and leptin in overweight and obese adults after the 12-week orange juice intervention [57]. This holds true not only in human intervention studies with fruit juice, but also with coffee. Coffee is a rich source of bioactive compounds, especially polyphenols such as phenolic acids, mostly chlorogenic or caffeic and in smaller amounts ferulic and p-coumaric acid but also alkaloids such as caffeine [58]. Bakuradze et al. revealed a significant decrease in body fat over the whole study period and a positive correlation between leptin and body fat of both female and male subjects [59].

In general, many studies have discussed the effects of polyphenols on blood lipid profiles [15,34,38]. A study with male healthy subjects showed a significant decrease in TC and LDL cholesterol within 24 h as well as after one-week intervention of an anthocyanin-rich fruit juice. However, these parameters increased towards the end of the study [9]. In line with these findings, in literature a meta-analysis indicated that daily supplementation for 6–8 weeks with aronia berry extracts significantly reduced TC with even larger effects among adults over the age of 50 years [60]. According to a study conducted by Habanova et al., regular consumption of bilberries (150 g of frozen stored bilberries three times a week for six weeks) decreased TC, LDL cholesterol, and TGs in healthy women (n = 25) and men (n = 11) [15]. Likewise, Koutsos et al. noticed that consumption of two apples daily (sum of polyphenols 990 mg) over eight weeks decreased TC, LDL cholesterol, and TGs in healthy mildly hypercholesterolemic volunteers [14]. In contrast, consumption of 750 mL cranberry juice over two weeks did not alter the lipid profile (TG, TC, HDL, and LDL cholesterol) measured in blood of 20 healthy female volunteers. The authors discussed that the lack of effect on plasma lipid profiles may reflect the relatively short duration of the study [61]. In the present study, we observed no changes of blood lipid profiles in both groups except significantly increased TG levels in placebo group during the whole intervention. Both, the study of Duthie et al. and our study were conducted with healthy subjects, and additionally the number of participants was low, which might have an influence on the lack of effect regarding blood lipid profile. Nevertheless, the correlation analysis showed a significant positive correlation of TG levels with body weight and BMI after four weeks of intervention. As described by Zaha et al., an increased BMI is associated with statistically significant higher waist and hip circumference, systolic and diastolic blood pressure, fasting glycemia or TGs [25]. Furthermore, lipidomics analysis showed that lipid profile of the participants changed during PRB consumption, and the saturation level of the fatty acid moieties decreased significantly (detailed information see Chipeaux et al., submitted). It was the first time lipidomics analysis of some of these compounds was conducted in the frame of a human intervention study. Due to limited knowledge about the correlation of lipid species and lipidomics, respectively, with blood lipids such as TGs, TC, HDL, and LDL cholesterol, future investigations are crucial.

In the present study, we investigated the PDE activity in platelets as another important parameter for monitoring the biological effects of PRB consumption on lipid metabolism. Especially the enzyme PDE 3B is essential for the regulation of pathways mediated by the second messenger cAMP and thus, can influence lipid metabolism via stimulation of lipolysis when PDE activity is inhibited [18]. There is evidence, that extracts rich in polyphenols from fruits or their juices can inhibit PDE activity in vitro [19,20,23]. Our previous investigations with extracts from red fruit juices and concentrates, their fractions (anthocyanin, copigment, and polymer) as well as selected anthocyanins on PDE 3B activity showed that the extract of chokeberry was a potent inhibitor of PDE 3B, followed by the extracts of blueberry, pomegranate, and cranberry. Additionally, the respective copigment fractions of chokeberry, cranberry, and pomegranate extracts were strong inhibitors of PDE 3B activity, too [21,29]. In this study, a significant inhibition of platelet cAMP-PDE activity was shown after PRB consumption, which was more pronounced compared to placebo group. In consideration of our in vitro data, the trends in PDE inhibition observed in this human study might be linked to the bioactive compounds of red fruits. Not much research has been conducted on the influence of polyphenols on PDE inhibition in vivo yet. Nevertheless, Montoya et al. investigated the modulation of cAMP-PDE activity in platelets after coffee consumption. They revealed a significantly reduced PDE activity after coffee intervention that was not directly dependent on the caffeine content of coffee. In addition, they had closer look at the effects of single coffee constituents in platelets isolated from platelet-rich plasma and described a similar PDE-inhibiting effect for the polyphenols 5-caffeoylquinic acid and caffeic acid as for caffeine [22].

In addition to the above mentioned parameters, the plasma uric acid concentrations were determined as indicator in the frame of clinical trials, since overproduction of uric acid can indicate the prevalence of the metabolic syndrome and can lead to the occurrence of gout [27,28]. It was shown that elevated uric acid levels (hyperuricemia) were significantly associated with visceral fat accumulation [62]. This effect might be mediated by intracellular and mitochondrial oxidative stress [63,64]. We observed slight decreases in plasma uric acid concentrations during intervention in both test group and placebo group. Previous studies have shown that high fruit and soybean intake for three months could be a way to reduce blood uric acid in asymptomatic hyperuricemia Chinese patients (n = 187) [65]. In line with these findings, Montmorency tart cherry concentrate (60 mL) significantly decreased serum urate with the greatest mean change at 8 h post-supplement in 20 healthy participants [66], which had been confirmed in a study by Hillman and Uhranowsky [67]. Büsing et al. investigated the effects of orange juice and caffeine-free soft drink (cola) consumption in 26 healthy adults in a 2 × 2-week intervention period and found that levels of uric acid did not change with soft drink (cola) uptake but decreased with orange juice intervention due to increased uric acid excretion. They discussed an uricosuric effect of vitamin C, since the mean intake of vitamin C from orange juice was 447 ± 78 mg d^−1^ [68]. Red fruit juices of chokeberry, cranberry, or pomegranate contained in the PRB have lower naturally occurring amounts of vitamin C than orange juice [69,70,71]. In addition, consumed foods can influence uric acid levels, since dietary fructose, meat and seafood consumption are associated with higher serum levels of uric acid (independent from protein intake), whereas dairy consumption was inversely associated with serum uric acid level [72,73]. Furthermore, we performed the present study with healthy young men with normal uric acid levels, so it is difficult to detect a preventive effect of PRB consumption. Correlation analysis showed a significant positive correlation for uric acid content with total DNA strand breaks in whole blood. Del Bo’ et al. found a positive correlation between the levels of H_2_O_2_-induced DNA damage and uric acid in women, which could not be observe in men [74]. Inconsistent effects had been described for uric acid as antioxidant. It is suggested that uric acid could play an important role as antioxidant and could be involved in protecting DNA from damage [75]. Despite its proposed protective properties, at the same time, high levels of uric acid were recognized as a marker of inflammatory states, cardiovascular, and metabolic diseases [76,77,78].

Many studies reported the antioxidant potential of polyphenols in vitro as wells as in vivo [5,8,79,80,81,82]. The PRB applied in the present study exhibited a high antioxidative capacity, reflected by a Trolox Equivalent Antioxidant Capacity (TEAC) value of 20.9 mmol L^−1^ Trolox compared to the placebo drink with 2.8 mmol L^−1^ Trolox. As we know, antioxidants can neutralize or protect a biological system from reactive oxygen species and possess DNA-protective activities. Recently published studies from our group regarding DNA protective effects of anthocyanin-rich fruit juice and supplements in humans showed a significant reduction in background and total DNA strand breaks as well as reduced H_2_O_2_-induced DNA strand breaks ex vivo [9,80]. Under healthy conditions background DNA damage is low and it is difficult to detect any potentially protective effects of PRB intervention. Thus, we used H_2_O_2_-treatment in isolated lymphocytes to induce an acute oxidative condition that mimics what may happen in diseased state. To verify the results from our previous research, we investigated the effects of PRB consumption on DNA strand breaks during this study. We could show that intervention of the beverage rich in polyphenols reduced total DNA strand breaks in both whole blood and lymphocytes treated with H_2_O_2_. This result is in accordance with data on DNA-protective effects of wild blueberry drink intake published by Riso et al., who reported that endogenously oxidized DNA bases and the levels of H_2_O_2_-induced DNA damage were significantly reduced in 18 male volunteers with cardiovascular risk factors after six weeks of intervention [83]. Fortunately, similar effects of red fruit juices and aronia juice-based food supplements on DNA integrity over an eight-week intervention period could be demonstrated in two studies with healthy men by our group [9,80]. Comparable results on DNA integrity by reduced spontaneous DNA strand breaks were obtained after consumption of other sources rich in polyphenols such as coffee [81].

In conclusion, this study demonstrated the beneficial effects of a PRB (containing chokeberry, cranberry, and pomegranate juice) on human health. The focus of the study was on parameters of the lipid metabolism, and we could observe an influence on reduced energy intake, which was correlated by a decrease in fat and protein intake. Furthermore, an inhibited PDE activity might be partly responsible for the stimulation of lipolysis and consequently, a modulation of body weight and composition. In addition, the beverage rich in polyphenols tended to improve DNA integrity. We are aware that our study is only a pilot intervention study since it has a limited number of participants. Future studies with an extended number of volunteers should be performed. Since we had the main focus in this study on the lipid metabolism, investigations with overweight or obese people should be performed to confirm the observed effects, respectively. Complementary, follow up studies could aim to focus more on single food compounds utilising controlled-feeding trials. To have a closer look at gender-specific effects, intervention studies with men and women could be conducted.

## 4. Materials and Methods

### 4.1. Chemicals

All chemicals used in the presented research were of analytical grade.

### 4.2. Study Design

The present intervention study was approved by the Local Ethics Committee of Rhineland-Palatinate, Mainz, Germany (No. 2020-15101). It consisted of a nine-week prospective, randomized, placebo-controlled human study with parallel design. Male volunteers (n = 41, BMI = 23.5 ± 1.8 kg m^−2^, age = 24.3 ± 2.3 years) who matched the following inclusion criteria were recruited to participate in the study: healthy, non-smokers, no practice of excessive sports, no intake of pharmaceutical drugs or food supplements during the study period, no pre-existing disease, not simultaneously participating in another study, and not donate blood during the observation period. Only male volunteers were selected in the study to have fewer hormonal fluctuations than in females. The volunteers were subjected to a standard medical health check including a questionnaire, blood pressure measurements, anthropometric measurements, and standard clinical blood biochemistry tests, after providing their informed written consent. The volunteers were then randomly divided into two groups according to their BMI (test group, n = 20, BMI = 23.2 ± 2.1 kg m^−2^; placebo group, n = 21, BMI = 23.7 ± 1.4 kg m^−2^). Five volunteers dropped out of the study for illness or private reasons. The remaining 36 volunteers (test group, n = 18; placebo group, n = 18) completed the nine-week study. After one-week washout period (no food rich in polyphenols), the volunteers consumed 750 mL of PRB or placebo drink assigned to their group in three equal portions over eight weeks (Figure 8). The volunteers were instructed to keep their normal dietary habits and lifestyle during the study, except the intake of foods rich in polyphenols such as red berries, coffee, or dark chocolate, and the consumption of foods rich in fat, which should have been avoided. At each examination appointment—after the washout period, after four and eight weeks of intervention—body weight as well as body composition of the study subjects were assessed on an empty bladder and both venous blood and spot urine samples were collected. Further, nutrient intake was recalled from seven-day dietary protocols recorded in the last week before each examination appointment.

### 4.3. Sample Preparation and Analysis of the Study Beverages

The study beverages (PRB and placebo drink) were produced at the Hochschule Geisenheim (Germany) and filled into in brown 750 mL bottles. The concentrates, juices and other materials were provided by Eckes-Granini Group GmbH (Niederolm, Germany), riha WeserGold GmbH & Co. KG (Rinteln, Germany), and ADM Wild Europe GmbH & Co. KG (Eppelheim, Germany). The PRB with 73% juice content was produced from water, rectified grape juice from concentrate (22%), chokeberry juice from concentrate (17%), cranberry juice from concentrate (17%), and pomegranate juice from concentrate (17%). The placebo drink was prepared from water, rectified grape juice from concentrate, citric and malic acids, and flavorings.

Soluble solids (Brix values) were measured refractometrically in accordance with the International Fruit and Vegetable Juice Association (IFU)-8 method (2017). For determination of myo-inositol as well as sorbitol ion chromatography with electrochemical detection was used, which was based on the IFU-79 method (2011). The sugar content of glucose and fructose was determined enzymatically with the method ASU L 31.00-12 (1997-01), citric acid with ASU L 31.00-14 (1997-01), and malic acid with ASU L 31.00-15 (1997-01).

Total polyphenol content was determined using the Folin-Ciocalteu-Assay and expressed as the concentration of GAE as previously reported [21].

The antioxidative capacity was determined using the TEAC assay according to method published by Re et al. [84]. This method is based on the decolorization of 2,2’-Azinobis-(3-ethylbenzothiazoline-6-sulfonic acid) (ABTS) radical cation. ABTS had been pre-activated by 2.45 mM potassium persulfate for at least 12 h in the dark and diluted with ethanol to reach an absorbance of 0.7–0.9 at 734 nm. For the measurement, 1 mL diluted ABTS^•+^ solution was put in semi-micro cuvettes and the absorbance (A1) was determined at 734 nm. After adding the diluted samples (1:50 or 1:100), calibration standards, positive control (ascorbic acid, *c* = 50 mg L^−1^), and solvent controls, the absorbance was measured again after 6 min (A2). Trolox was used for calibration (final concentrations 0.1–1.5 mM).

Identification of the anthocyanins and copigments by HPLC-ESI-MS/MS as well as quantification by ultra-high-performance liquid chromatography-diode array detector (UHPLC-DAD) was performed as described before by Niesen et al. [29].

### 4.4. Nutrient Intake

The nutrient intake (kcal, carbohydrates, fat, and proteins) of each subject during each study period was calculated on the basis of the seven-day food and beverage records using the nutrition software package PRODI 5 Expert (Nutri-Science GmbH, Hausach, Germany).

### 4.5. Anthropometric Measurements

Anthropometric measurements were performed in the morning before breakfast with an emptied bladder as follows: the subjects’ body weights were determined using a medical scale (Seca delta 707; Seca, Hamburg, Germany) and their heights were measured using a Seca 206 measuring tape (Seca) to calculate their BMI (kg m^−2^). A bioelectrical impedance analyzer 101 (BIA 101, SMT medical GmbH, Würzburg, Germany) was used to estimate body composition (total body water (TBW), FM, and FFM). The measurements were performed at a horizontal position. Special skin electrodes were placed on the right hand and foot on dry skin according to the manufacturer’s instructions.

### 4.6. Blood and Urine Collection, Processing, and Storage

Venous blood was collected in EDTA tubes (Sarstedt, Nümbrecht, Germany) and centrifuged at 2500× *g* for 10 min at room temperature. The achieved plasma was used for the quantification of GLP-1 and leptin as well as the concentration of uric acid. For blood lipid measurements (TG, TC, HDL cholesterol, LDL cholesterol) and lipase activity, the venous blood was collected in Li-heparin tubes and sent immediately to the Department of Laboratory Medicine (Westpfalz-Klinikum Kaiserslautern, Germany) for analysis. For cAMP-PDE activity assay, platelets were isolated from platelet-rich plasma by centrifugation (150× *g*, 15 min, 24 °C). The platelet-rich plasma was once more centrifuged at 800× *g* for 5 min at 24 °C. Supernatants were removed and pellets were gently resuspended in Tris buffer. To assess the amount of DNA strand breaks, the blood samples in EDTA tubes were immediately processed for the comet assay. Human PBLs were isolated by centrifugation with Histopaque-1077 (Sigma-Aldrich, Steinheim, Germany). Briefly, 7 mL of freshly collected human blood was layered onto 7 mL of Histopaque-1077 and continuously centrifuged at room temperature at 400× *g* for 25 min. PBLs were collected from the layer between the plasma and Histopaque 1077, transferred into 10 mL RPMI 1640 medium (Gibco, Life Technologies, Darmstadt, Germany) supplemented with 10% fetal calf serum and 1% penicillin/streptomycin, and tempered at 37 °C. Thereafter, the cell suspension was centrifuged for 10 min (250× *g*), and the pellet was resolved in 6 mL RPMI 1640 medium and repeatedly centrifuged. Immediately afterwards, the PBLs were analyzed using the comet assay. For the stabilization of anthocyanins in urine, the collected spot urine samples were adjusted to pH 2.5 with hydrochloric acid (1 M). All samples were stored in aliquots at −80 °C until analysis was performed.

### 4.7. Blood Lipids and Lipase Activity

The blood lipids such as TG, TC, HDL cholesterol, LDL cholesterol, and lipase activity were determined with standard methods at the Department of Laboratory Medicine (Westpfalz-Klinikum Kaiserslautern, Germany).

### 4.8. Human Blood Platelet cAMP-PDE Activity Assay

For cAMP-PDE activity assays, isolated platelets were resuspended in Tris buffer (50 mM Tris/HCl, pH 7.4, 10 mM MgCl_2_, 0.1 mM EDTA, 0.5 mM β-mercaptoethanol, 5 mM benzamidine, and a protease inhibitor mix) and lyzed by ultrasonic treatment on ice. After centrifugation (12,000× *g*, 15 min, 4 °C), the supernatant was removed and directly used for the PDE assay. PDE activity was determined according to a method by Pöch and Montoya et al. with slight modifications [22,85]. Briefly, 50 µL of the samples and 50 µL of the Tris buffer were incubated at 37 °C with 50 µL cAMP mix (30 mM Tris/HCl, pH 7.4, 9 mM MgCl_2_, 3 mM 5′-AMP, 3 μM cAMP, and 96.2 kBq mL^−1^ [2,8-^3^H]-cAMP). The enzyme reaction was stopped on ice after 20 min with 250 µL ZnSO_4_ (0.266 mM) and [2,8-^3^H]-5′-AMP was precipitated by adding 250 µL Ba(OH)_2_ (0.266 mM). Tubes were centrifuged (13,000× *g*, 5 min, 4 °C) and then 450 µL of the supernatant was mixed with 3.5 mL of a scintillation cocktail, and the resulting radioactivity was measured using a liquid scintillation counter (Wallac 1410, Pharmacia, Uppsala, Sweden). Typing for PDE isoenzyme activity in platelet-rich plasma from the intervention study subjects was achieved with the aid of specific inhibitors, namely 50 µM erythro-9-(2-hydroxy-3-nonyl)adenine (EHNA; PDE 2), 10 µM Milrinone (PDE 3), 50 µM Rolipram (PDE 4), and 10 µM Zaprinast (PDE 5) [22,32]. For each blood sample, three independent measurements were performed. Data were normalized to protein content and expressed in units of PDE activity (pmol cAMP min^−1^ mg^−1^ protein). Therefore, protein concentrations of the samples were determined according to the method of Bradford using bovine serum albumin as a standard [86].

### 4.9. Leptin and GLP-1

Plasma leptin and GLP-1 levels were determined using commercially available sandwich enzyme-linked immunosorbent assay (ELISA) kits (Thermo Fisher Scientific, Vienna, Austria). All analyses were carried out according to the manufacturer’s protocols.

### 4.10. Uric Acid

The uric acid content in the plasma samples was measured using HPLC-ESI-MS/MS with isotope labelled 1,3-^15^N_2_-uric acid as an internal standard (IS) according to the method published by Kim et al. with slight modifications [87]. The stock solutions were prepared in 0.3 M KOH at a concentration of 1500 µg mL^−1^ for uric acid and 1600 µg mL^−1^ for IS and stored at −18 °C. The quantification of uric acid in plasma was carried out by a calibration curve using concentrations from 2 µg mL^−1^ to 100 µg mL^−1^, with a final concentration of 40 µg mL^−1^ IS. The calibration curves were plotted as concentration vs. peak area ratio of the analytes and the IS, yielding a correlation coefficient of R^2^ > 0.99. Signal-to-noise ratios were defined as 3:1 relative to the LOD and 10:1 relative to the LOQ. LOD was determined as 0.2 µg mL^−1^ and LOQ as 0.4 µg mL^−1^. Intra-day repeatability was assessed by ten replicate analyses of one standard concentration plus fixed concentration of IS. Inter-day reproducibility was obtained by analysis of one concentration level of the analyte plus fixed concentration of IS for five days in a row. The coefficient of variation was 3.4% for intra-day and 4.6% for inter-day experiments. Recovery of 94 ± 8% was determined by spiking plasma with 8, 10, and 12 µg mL^−1^ of uric acid. The IS (40 µg mL^−1^) was added to 200 µL plasma sample. After 3 h equilibration at 4 °C, 40 μL of trichloroacetic acid (10%) was added for protein precipitation and the sample was centrifuged for 1 min at 20 °C and 17,000× *g*. The supernatant was filtered through a 0.2 µm nylon filter by centrifugation for 5 min at 17,000× *g*. The filtrate was further analyzed by HPLC-ESI-MS/MS.

Analysis of the samples was performed with an HPLC Agilent 1100 Series equipped with a degasser (G1322A), quaternary pump (G1311A), autosampler (G1313A), column oven (G1316A) (Agilent Technologies, Santa Clara, CA, USA) and coupled an API 2000 triple-quadrupole mass spectrometer (Applied Biosystems, Framingham, MA, USA) with ESI source. Liquid chromatography analyses were performed in a gradient elution mode using Phenomenex Luna 5 μm C18(2) 100Å (250 mm × 4.6 mm) column (Phenomenex, Aschaffenburg, Germany) coupled with a Phenomenex Luna C8(2), 5 μm particle size guard column. The mobile phase used included 5 mM ammonium acetate/0.1% acetic acid (A) and acetonitrile/0.1% formic acid (B). The mobile phase flow was 0.9 mL min^−1^. The injection volume was 20 μL. The gradient was as follows: 0–3.8 min, 5% B; 3.8–6.7 min, 5–35% B; 6.7–7.6 min, 35%B; 7.6–8.0 min, 35–95% B; 8.0–9.8 min, 95% B; next, reversed to the original composition of 5% B over 0.5 min, after which it was kept constant for 1.8 min to re-equilibrate the column. The ESI source was operated in negative ion mode (−4000 V), and nitrogen was used as nebulizer gas (45 psi), heater gas (30 psi, 500 °C), curtain gas (30 psi), and collision gas (4 psi). Uric acid as well as the isotopically labelled IS were detected in multiple reaction monitoring (MRM) mode. The parent ion of uric acid was *m*/*z* 167.0 and monitored MRM ions were *m*/*z* 123.9 and 95.8. In case of IS the parent ion of 1,3-^15^N_2_-uric acid was *m*/*z* 169.0 and monitored MRM ions were *m*/*z* 124.9 and 96.9. The substance-specific parameters of individual compounds are listed in Table 6.

### 4.11. Comet Assay in Whole Blood Samples and in Isolated Human Peripheral Blood Lymphocytes (PBLs)

Alkaline single-cell gel electrophoresis (comet assay in whole blood with and without FPG solution) was performed according to Collins et al. [88] with slight modifications as previously reported [89].

For the comet assay in PBLs, 2 × 50,000 freshly isolated PBLs were centrifuged, and the pellet was mixed with low-melting agarose. Next, the slides were exposed to H_2_O_2_ (50 µM) on ice for 5 min, according to previous publications [80]. After the treatment, the cells were subjected to lysis, washed, then incubated in 50 µL FPG solution for 30 min at 37 °C. After DNA unwinding and horizontal gel electrophoresis, slides were washed, stained with GelRed, and analyzed using a fluorescence microscope (Imager, A1, filter set 15, Zeiss, Germany) and computerized image analysis (Comet IV, Perceptive Instruments), scoring 2 × 50 images per slide (2 gels per slide). DNA migration was directly expressed as mean tail intensity (%) from two gels.

### 4.12. Compliance Analysis

Compliance to the study was ascertained by monitoring urinary excretion of anthocyanins in all volunteers. The anthocyanin content of acidified spot urine samples was determined according to Mueller et al. with slight modifications [30]. Urine samples (6 mL), which included 20 µL of delphinidin-3,5-diglucoside (100 µg mL^−1^), were applied to a solid-phase extraction cartridge (Strata C18, 500 mg, Phenomenex, Aschaffenburg, Germany), preconditioned with methanol (4 mL) and equilibrated with water/formic acid (95/5, *v*/*v*, 4 mL). Elution was performed with methanol/formic acid (95/5, *v*/*v*, 3 mL), eluates were concentrated using a vacuum centrifuge (Eppendorf, Hamburg, Germany). The residue was resuspended in 200 µL of water/formic acid/acetonitrile (87/10/3, *v*/*v*/*v*). After centrifugation (16,000× *g*, 5 min), 20 µL of the supernatant was analyzed with HPLC-ESI-MS/MS using an Agilent HPLC system (Agilent, Santa Clara, CA, USA) coupled to an API 3200 MS (AB Sciex, Framingham, MA, USA). HPLC chromatographic separation was carried out with a Luna 5 µm C18(2) 100Å (250 mm × 4.6 mm) column (Phenomenex, Aschaffenburg, Germany) coupled with a Phenomenex Luna C18(2), 5 μm particle size guard column. The HPLC parameters were as follows: flow 0.7 mL min^−1^; solvent A (water/formic acid/acetonitrile, 87/10/3, *v*/*v*/*v*); solvent B (acetonitrile/water/formic acid, 50/40/10, *v*/*v*/*v*). The gradient was comprised of 0–0.7 min, 2% B; 0.7–14.9 min, 2–14% B; 14.9–28.6 min, 14% B; 28.6–35.7 min, 14–15% B; 35.7–39.3 min, 15–19% B; 39.3–46.4 min, 19–20% B; 46.4–46.5 min, 20–99% B; linear step to 99% B for 4.5 min; and return to initial condition of 2% B for 0.1 min (equilibration time: 16.9 min). The ESI-MS instrument-parameters were carried out in positive mode: curtain gas (30 psi), ion spray (5000 V), temperature (450 °C), gas 1 (50 psi), and gas 2 (40 psi). The substance-specific parameters of individual compounds and transitions are listed in Table 7. For identification, a standard mix of the IS (delphinidin-3,5-diglucoside) and the two main anthocyanins of the PRB, namely cyanidin-3-galactoside and cyanidin-3-arabinoside, was measured at a final concentration of 1 µg mL^−1^. Cyanidin-3-galactoside was used as qualifier and a LOD of 0.05 µg mL^−1^ was determined.

### 4.13. Statistical Analysis

The results of tested parameters were presented as the mean and standard deviation (SD). Statistical analyses were conducted using the Analysis Tool Excel of Microsoft 365 Apps for Enterprise (Microsoft Corporation, Redmond, WA, USA), Origin 2020 (OriginLab, Northampton, MA, USA) and R 2.13.1 (R Foundation for Statistical Computing, Vienna, Austria). The Anderson-Darling test was used for the analysis of normal distribution. The statistical significance of differences in parameters between the study phases within each respective group were analyzed with a one-sided paired t-test (differences normally distributed) or a one-sided paired Wilcoxon’s signed rank test (differences without normal distribution). The statistical significance of differences in parameters between the test group and placebo group were analyzed by either a two-sample t-test (normal distribution) or a Mann-Whitney U-test (non-normal distribution). Spearman correlation coefficients were calculated based on the changes of the washout period to both the intervention period after four and after eight weeks. Significant correlations were determined by means of t-test.

## Figures and Tables

**Figure 1 ijms-24-00841-f001:**
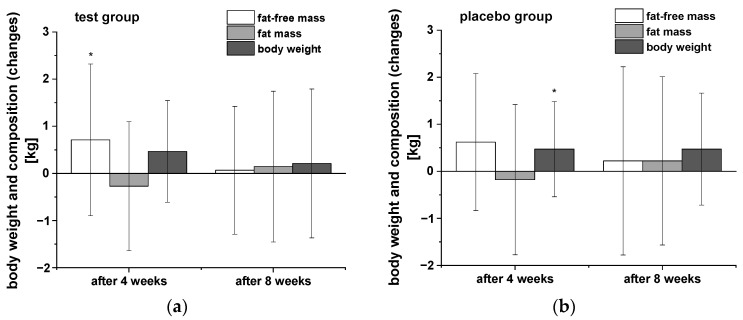
The changes in body weight, fat-free mass, and body fat of participants in both groups during the intervention in comparison to the washout period: (**a**) test group (n = 18), (**b**) placebo group (n = 18). Data are presented as mean values and SD of differences. Significant differences in body weight and composition relative to the washout period: * *p* < 0.05.

**Figure 2 ijms-24-00841-f002:**
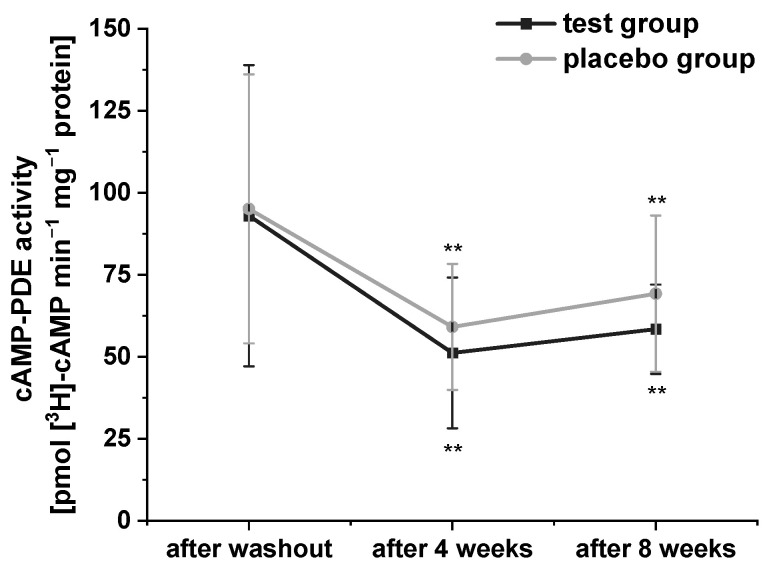
The modulation of cyclic adenosine monophosphate (cAMP)-phosphodiesterase (PDE) activity in the platelets of subjects. Values are expressed as means ± SD. Data were normalized to protein content and expressed in units of cAMP-PDE activity (pmol cAMP min^−1^ mg^−1^ protein). Significant differences in comparison to the washout period: ** *p* < 0.01.

**Figure 3 ijms-24-00841-f003:**
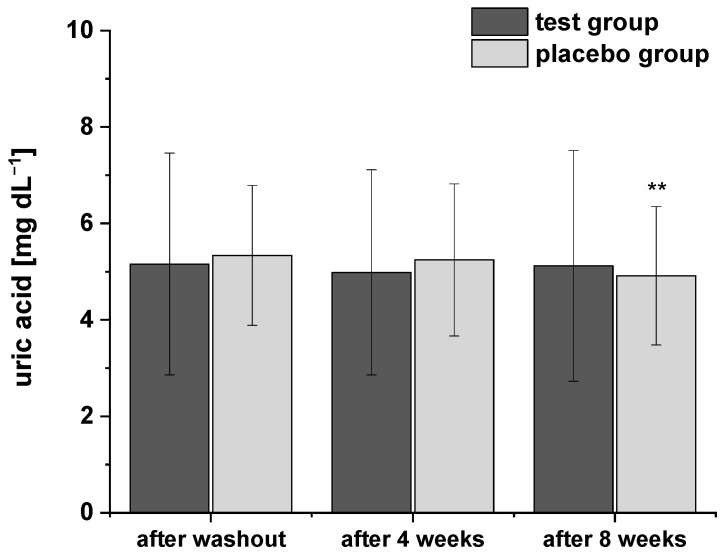
The plasma uric acid concentrations after the washout period, after four and eight weeks of intervention. Data are expressed as means ± SD. Significant differences in comparison to the washout period: ** *p* < 0.01.

**Figure 4 ijms-24-00841-f004:**
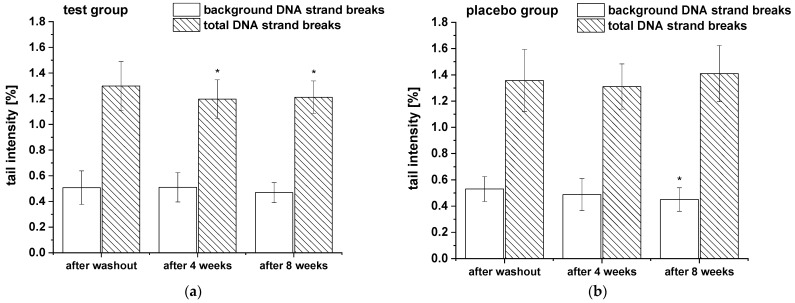
The background and total DNA strand breaks in whole blood during the nine-week study: (**a**) test group (n = 18), (**b**) placebo group (n = 18). Data are expressed as mean tail intensity [%] ± SD. Significant differences in comparison to the washout period: * *p* < 0.05.

**Figure 5 ijms-24-00841-f005:**
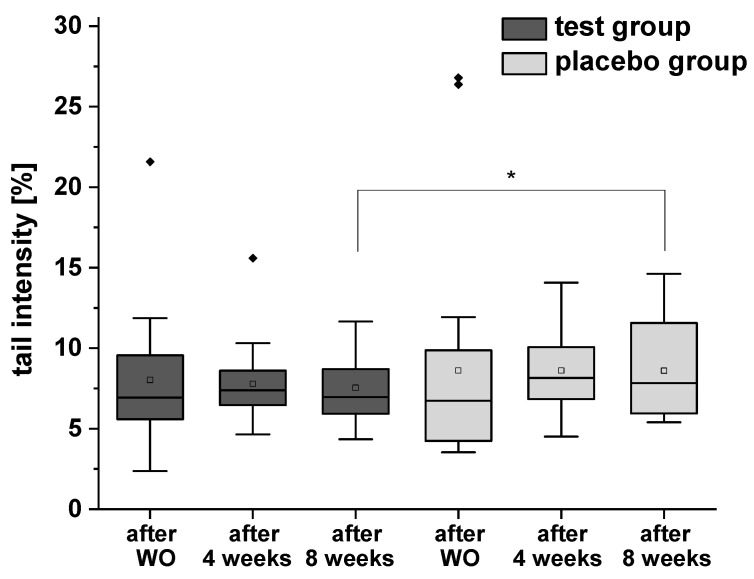
H_2_O_2_-induced DNA strand breaks (with formamidopyrimidine-DNA glycosylase (FPG) treatment) in isolated peripheral blood lymphocytes after the washout (WO) period, after four and eight weeks of intervention. Data are expressed as mean tail intensity [%] ± SD. Squares represent mean values; horizontal lines within the boxes represent median values; rhombuses represent outliers. * *p* < 0.05.

**Figure 6 ijms-24-00841-f006:**
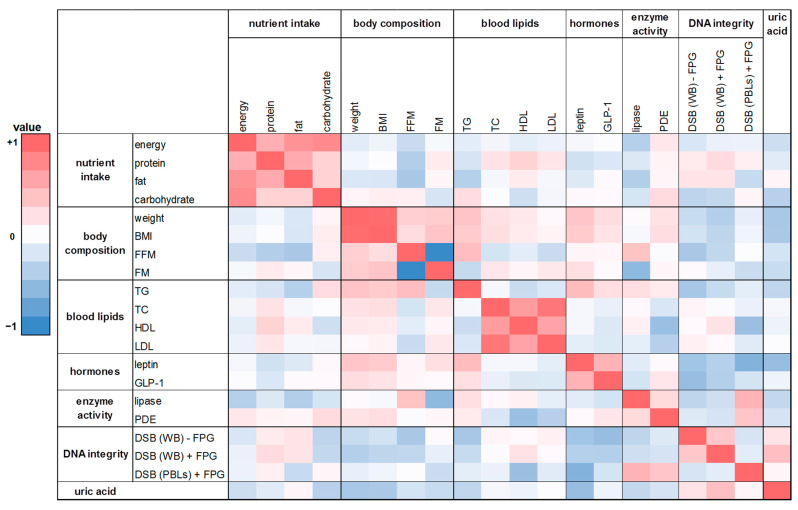
The heat map correlation matrix of all parameters investigated during the human intervention study. The coefficients were calculated based on the changes between the washout period and the intervention period after four weeks. Colored fields: red: positive correlation; blue: negative correlation. BMI: body mass index; DSB: DNA strand breaks; TC: total cholesterol; TG: triglyceride; WB: whole blood.

**Figure 7 ijms-24-00841-f007:**
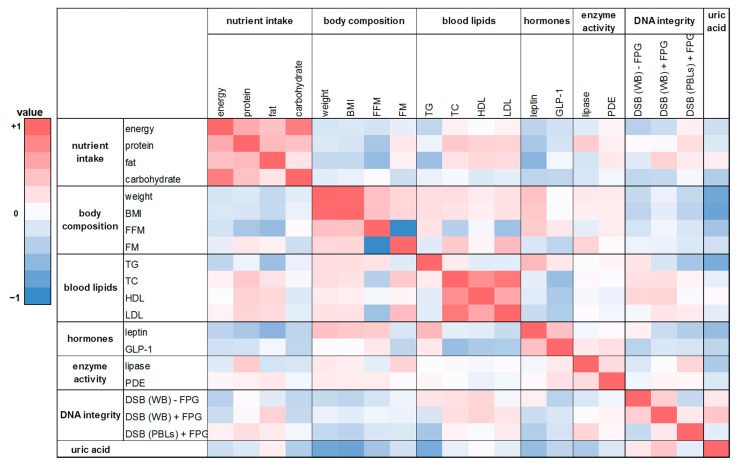
The heat map correlation matrix of all parameters investigated during the human intervention study. The coefficients were calculated based on the changes between the washout period and the intervention period after eight weeks. Colored fields: red: positive correlation; blue: negative correlation.

**Figure 8 ijms-24-00841-f008:**
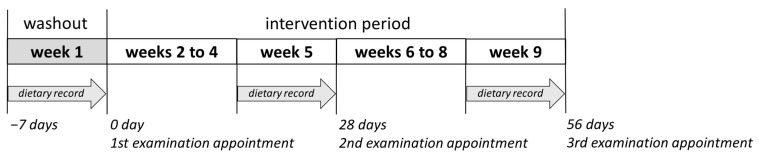
The design of the nine-week placebo-controlled intervention study.

**Table 1 ijms-24-00841-t001:** The composition of the two study beverages.

	Polyphenol-Rich Beverage	Placebo Drink
brix [°]	9.2	8.3
glucose [g L^−1^]	35.3	35.2
fructose [g L^−1^]	33.3	38.6
myo-inositol [g L^−1^]	0.2	0.1
sorbitol [g L^−1^]	8.67	n.d.
citric acid [g L^−1^]	3.7	3.7
malic acid [g L^−1^]	2.8	2.8
total phenolic content (Folin- Ciocalteu assay) [g GAE L^−1^]	3.0	0.1
antioxidative capacity (TEAC assay) [mmol L^−1^]	20.9	2.8
total anthocyanins [mg L^−1^]	735.8	n.d.
total flavonoids [mg L^−1^]	235.0	n.d.
total chlorogenic acids [mg L^−1^]	306.9	n.d.
total tannins [mg L^−1^]	400.8	n.d.
polymeric content [mg L^−1^]	367.1	n.d.

GAE: gallic acid equivalents; n.d.: not detectable; TEAC: Trolox Equivalent Antioxidant Capacity.

**Table 2 ijms-24-00841-t002:** The average daily energy and nutrient intakes of volunteers calculated with PRODI 5 Expert software, based on seven-day food records completed in the one week of the washout period, four and eight weeks of intervention. Data are presented as means and standard deviation (SD); significant differences in comparison to the washout period: * *p* < 0.05; ** *p* < 0.01.

	After Washout Period	After Four Weeks of Intervention	After Eight Weeks of Intervention
**test group**			
energy intake [kcal] protein [g] fat [g] carbohydrates [g]	2142.5 ± 494.5 87.5 ± 22.2 77.7 ± 19.6 239.6 ± 51.5	2125.3 ± 358.9 77.1 ± 12.7 * 77.5 ± 19.6 263.4 ± 64.0	1999.2 ± 418.0 70.3 ± 22.7 ** 67.2 ± 27.9 * 245.9 ± 66.9
**placebo group**			
energy intake [kcal] protein [g] fat [g] carbohydrates [g]	2139.1 ± 663.3 85.9 ± 27.8 79.8 ± 31.8 259.1 ± 114.3	2125.7 ± 629.1 82.4 ± 32.4 80.6 ± 34.0 256.5 ± 74.0	2116.6 ± 682.3 83.1 ± 37.6 78.5 ± 30.2 252.5 ± 81.1

**Table 3 ijms-24-00841-t003:** Body weight and body composition of the participants throughout the study. Data are presented as mean values and SD. Significant differences in comparison to the washout period: * *p* < 0.05.

	After Washout Period	After Four Weeks of Intervention	After Eight Weeks of Intervention
**test group**			
body weight [kg]	75.5 ± 8.9	75.9 ± 8.2	75.7 ± 8.5
FFM [kg]	59.3 ± 5.9	60.0 ± 6.7 *	59.4 ± 6.0
FM [kg]	16.2 ± 4.8	15.9 ± 4.3	16.3 ± 4.3
TBW [L]	43.4 ± 4.3	43.9 ± 4.9	43.4 ± 4.4
**placebo group**			
body weight [kg]	79.4 ± 7.9	79.9 ± 7.8 *	79.9 ± 7.7
FFM [kg]	61.7 ± 6.1	62.3 ± 5.7	61.9 ± 5.5
FM [kg]	17.7 ± 4.3	17.6 ± 4.3	18.0 ± 4.2
TBW [L]	45.2 ± 4.4	45.6 ± 4.2	45.3 ± 4.0

FFM: fat-free mass; FM: fat mass; TBW: total body water.

**Table 4 ijms-24-00841-t004:** The blood lipid profiles and lipase activity of the participants throughout the study. Data are presented as mean values and SD. Significant differences in comparison to the washout period: ** *p* < 0.01.

	After Washout Period	After Four Weeks of Intervention	After Eight Weeks of Intervention
**test group**			
triglycerides [mg dL^−1^]	87.2 ± 53.4	84.8 ± 27.1	94.8 ± 43.8
total cholesterol [mg dL^−1^]	171.4 ± 20.3	172.2 ± 16.1	170.3 ± 21.9
HDL cholesterol [mg dL^−1^]	54.6 ± 11.6	54.9 ± 11.1	49.9 ± 10.0
LDL cholesterol [mg dL^−1^]	110.3 ± 19.2	112.8 ± 20.1	109.1 ± 19.3
lipase activity [U L^−1^]	31.3 ± 6.5	31.4 ± 6.4	31.7 ± 6.1
**placebo group**			
triglycerides [mg dL^−1^]	83.2 ± 30.4	102.3 ± 37.5 **	104.9 ± 41.3 **
total cholesterol [mg dL^−1^]	182.6 ± 30.7	181.5 ± 30.9	179.4 ± 36.9
HDL cholesterol [mg dL^−1^]	53.5 ± 10.8	54.3 ± 12.3	49.3 ± 14.0
LDL cholesterol [mg dL^−1^]	123.3 ± 29.6	120.7 ± 29.2	120.3 ± 28.5
lipase activity [U L^−1^]	39.1 ± 17.7	43.6 ± 29.8	43.6 ± 14.2

HDL: high-density lipoprotein; LDL: low-density lipoprotein.

**Table 5 ijms-24-00841-t005:** The mean plasma leptin and glucagon-like peptide-1 (GLP-1) concentrations after the washout period, as well as after four- and eight-weeks intervention period of 750 mL polyphenol-rich beverage or placebo drink. The values are shown as means ± SD. Significant differences in comparison to the washout period: * *p* < 0.05.

	After Washout Period	After Four Weeks of Intervention	After Eight Weeks of Intervention
**test group**			
leptin [pg mL^−1^]	617.8 ± 595.8	600.7 ± 494.2	616.2 ± 562.6
GLP-1 [pg mL^−1^]	19.3 ± 10.0	19.6 ± 8.3	15.0 ± 6.0 *
**placebo group**			
leptin [pg mL^−1^]	1088.4 ± 2332.7	1362.9 ± 2708.3 *	1230.7 ± 2379.5 *
GLP-1 [pg mL^−1^]	18.6 ± 10.3	19.2 ± 10.8	20.6 ± 10.6

**Table 6 ijms-24-00841-t006:** HPLC-ESI-MS/MS (MRM) ion transitions [*m*/*z*], retention times [min], and substance-specific parameters (DP: declustering potential, EP: entrance potential, FP: focusing potential, CEP: cell entrance potential, CE: collision energy, CXP: cell exit potential [V]) for uric acid and the isotope labelled 1,3-^15^N_2_-uric acid as internal standard (IS).

Compound	MRM Ion Transition [*m*/*z*]	Retention Time [min]	DP [V]	EP [V]	FP [V]	CEP [V]	CE [V]	CXP [V]
uric acid	167.0/123.9 167.0/95.8	4.1	−36.0 −36.0	−6.0 −6.0	−220.0 −220.0	−12.0 −12.0	−18.0 −22.0	−8.0 −6.0
IS (1,3-^15^N_2_-uric acid)	169.0/124.9 169.0/96.9	4.1	−36.0 −36.0	−11.0 −11.0	−230.0 −230.0	−12.0 −12.0	−20.0 −6.0	−8.0 −6.0

**Table 7 ijms-24-00841-t007:** HPLC-ESI-MS/MS (MRM) ion transitions [*m*/*z*], retention times [min], and substance-specific parameters (DP: declustering potential, EP: entrance potential, CEP: cell entrance potential, CE: collision energy, CXP: cell exit potential [V]) of anthocyanins.

Analyte	MRM Ion Transition [*m*/*z*]	Retention Time [min]	DP [V]	EP [V]	CEP [V]	CE [V]	CXP [V]
delphinidin-3,5-di- glucoside (IS)	627/303	12.5	76.0	9.5	29.4	49.0	26.0
cyanidin-3-galactoside	449/287	18.5	66.0	7.5	20.0	31.0	26.0
cyanidin-3-arabinoside	419/287	22.3	66.0	10.5	20.0	33.0	24.0

IS: internal standard.

## Data Availability

Data are contained within the article.

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
