# Peer review of "Polyphenol-Rich Beverage Consumption Affecting Parameters of the Lipid Metabolism in Healthy Subjects"

_ijms, 2023, doi:10.3390/ijms24010841_

Round 1
Reviewer 1 Report
ijms-2082150
Beneficial Health Effects of Polyphenol-Rich Beverage Consumption in Humans
Studying the effects of various bioactive compounds on the body with volunteers is a challenge because it is difficult to minimize the influence of various random factors. The work is interesting and I appreciate the knowledge and efforts of the Authors, however, I have a few comments regarding the design of this experiment.
It is not clear why the Authors used a placebo drink with a higher content of fructose in their studies on lipid metabolism. Fructose, as it is directly involved in the fatty acid synthesis pathway, will influence the final fatty acid content. The fructose concentrations in both drinks should have been equal. The difference is not big but the Authors should comment on this in the discussion.
Many people who care about their health, even without knowledge at the academic level, know that some polyphenols are colorful. If volunteers were told they were taking part in a study on the effects of polyphenols and were given a colorless drink, they could assume it was a placebo. Such a situation would be inconsistent with the purpose of using a placebo. How did the Authors solve this problem?
The results, discussion and description of the methodology have been prepared very carefully, with great attention to detail, showing extensive knowledge in the subject and using professional approaches. The text is well written and contains almost no editorial errors. The Authors should ensure that the units in square brackets will not be split into two lines, as they are now (Table 4, column 1).
Dec. 9, 2022
Author Response
- It is not clear why the Authors used a placebo drink with a higher content of fructose in their studies on lipid metabolism. Fructose, as it is directly involved in the fatty acid synthesis pathway, will influence the final fatty acid content. The fructose concentrations in both drinks should have been equal. The difference is not big but the Authors should comment on this in the discussion.
Response: Thank you for this comment. For the production of placebo drink, the concentrations of glucose and fructose were adjusted to the concentrations of the polyphenol-rich beverage. The reason was to obtain an isocaloric drink with the same composition as the polyphenol-rich beverage in our study. Nevertheless, the two study drinks differed slightly in their fructose content (33.3 g L-1 vs. 38.6 g L-1). We have added the information in the discussion section (see lines 301–303): ‘Fructose can have a lipogenic effect by stimulating the fatty acid synthesis [33]’.
- Many people who care about their health, even without knowledge at the academic level, know that some polyphenols are colorful. If volunteers were told they were taking part in a study on the effects of polyphenols and were given a colorless drink, they could assume it was a placebo. Such a situation would be inconsistent with the purpose of using a placebo. How did the Authors solve this problem?
Response: The participants received the drinks in dark brown glass bottles. As we could not exclude that a natural food color has an impact on lipid metabolism, so the placebo drink was produced without colorants.
- The Authors should ensure that the units in square brackets will not be split into two lines, as they are now (Table 4, column 1).
Response: Thanks! We have checked the manuscript that the units are not being split into two lines.
Reviewer 2 Report
Manuscript International Journal of Molecular Sciences-2082150 entitled " Beneficial Health Effects of Polyphenol-Rich Beverage Consumption in Humans" by Celina Rahn et al. is a research article and its aim is to investigate effects of polyphenol- rich beverage uptake on lipid metabolism as well as DNA integrity. The manuscript is similar to a paper published in 2019, an intervention study on the effects of consuming 750 ml of anthocyanin-rich fruit juice (ref. 9)
Specific comments.
The title needs to be changed. Title is too generic; the proposed title is suitable for a review.
The text is clear.
The methods are adequately described. Explain why only male subjects were selected
The results are clearly presented.
Rewrite the conclusions. For example: line 494 “In conclusion, this study demonstrated the beneficial effects of a PRB (containing chokeberry, cranberry, and pomegranate juice) on human health”. It is an incorrect, exaggerated statement, considering the small number of subjects analyzed (n=18). The authors must emphasize that this is a pilot study for the small number of subjects.
References are up to date and complete.
Author Response
- The title needs to be changed. Title is too generic; the proposed title is suitable for a review.
Response: The reviewer is right. We have changed the title to ‘Polyphenol-rich beverage consumption affecting parameters of the lipid metabolism in healthy subjects’.
- Explain why only male subjects were selected.
Response: The study was inspired by the intervention study of Bakuradze et al., (2019) with male healthy volunteers. During this study we got first insights on the effectiveness of polyphenol-rich beverage on lipid metabolism. Therefore, we also selected only men for the present study. The other reason was to avoid hormonal fluctuations, which have influence on body composition as well as on lipid metabolism. The next step would be an intervention study with both, female and male volunteers. We have added in section 4.2 (Study design). ‘Only male volunteers were selected in the study to have less hormonal fluctuations than in females’.
- Rewrite the conclusions. For example: line 494 “In conclusion, this study demonstrated the beneficial effects of a PRB (containing chokeberry, cranberry, and pomegranate juice) on human health”. It is an incorrect, exaggerated statement, considering the small number of subjects analyzed (n=18). The authors must emphasize that this is a pilot study for the small number of subjects.
Response: We thank the reviewer for this comment. We have added in the conclusion ‘We are aware that our study is only a pilot intervention study since it has a limited number of participants’.